# Mismatch Repair Deficiency as a Predictive and Prognostic Biomarker in Endometrial Cancer: A Review on Immunohistochemistry Staining Patterns and Clinical Implications

**DOI:** 10.3390/ijms25021056

**Published:** 2024-01-15

**Authors:** Francesca Addante, Antonio d’Amati, Angela Santoro, Giuseppe Angelico, Frediano Inzani, Damiano Arciuolo, Antonio Travaglino, Antonio Raffone, Nicoletta D’Alessandris, Giulia Scaglione, Michele Valente, Giordana Tinnirello, Stefania Sfregola, Belen Padial Urtueta, Alessia Piermattei, Federica Cianfrini, Antonino Mulè, Emma Bragantini, Gian Franco Zannoni

**Affiliations:** 1Unità di Ginecopatologia e Patologia Mammaria, Dipartimento Scienze della Salute della Donna, del Bambino e di Sanità Pubblica, Fondazione Policlinico Universitario A. Gemelli-IRCCS, Largo A. Gemelli 8, 00168 Rome, Italyantonio.damati@uniba.it (A.d.); giulia.scaglione@policlinicogemelli.it (G.S.); alessia.piermattei@policlinicogemelli.it (A.P.); gianfranco.zannoni@unicatt.it (G.F.Z.); 2Unit of Anatomical Pathology, Department of Precision and Regenerative Medicine and Ionian Area, University of Bari “Aldo Moro”, Piazza Giulio Cesare 11, 70124 Bari, Italy; 3Unit of Human Anatomy and Histology, Department of Translational Biomedicine and Neuroscience (DiBraiN), University of Bari “Aldo Moro”, Piazza Giulio Cesare 11, 70124 Bari, Italy; 4Istituto di Anatomia Patologica, Università Cattolica del Sacro Cuore, Largo A. Gemelli 8, 00168 Rome, Italy; 5Department of Medical and Surgical Sciences and Advanced Technologies “G. F. Ingrassia”, Anatomic Pathology, University of Catania, 95123 Catania, Italy; giuangel86@hotmail.it (G.A.);; 6Anatomic Pathology Unit, Department of Molecular Medicine, University of Pavia and Fondazione IRCCS San Matteo Hospital, 27100 Pavia, Italy; frediano.inzani@unipv.it; 7Pathology Unit, Department of Medicine and Technological Innovation, University of Insubria, 21100 Varese, Italy; 8Gynecology and Obstetrics Unit, Department of Public Health, University of Naples Federico II, 80138 Naples, Italy; 9Department of Surgical Pathology, Ospedale S. Chiara, Largo Medaglie d’Oro 9, 38122 Trento, Italy

**Keywords:** MSI, MMR deficiency, endometrial cancer, histomolecular prognostic risk assessment, immunotherapy

## Abstract

Among the four endometrial cancer (EC) TCGA molecular groups, the MSI/hypermutated group represents an important percentage of tumors (30%), including different histotypes, and generally confers an intermediate prognosis for affected women, also providing new immunotherapeutic strategies. Immunohistochemistry for MMR proteins (MLH1, MSH2, MSH6 and PMS2) has become the optimal diagnostic MSI surrogate worldwide. This review aims to provide state-of-the-art knowledge on MMR deficiency/MSI in EC and to clarify the pathological assessment, interpretation pitfalls and reporting of MMR status.

## 1. Introduction

Endometrial carcinoma (EC) represents the most common gynecological malignancy in Europe and USA, with a frequency of 15 to 25 per 100,000 women in Western countries [1,2]. In most of the cases, patients present an early-stage tumor at diagnosis, with an excellent prognosis. In 15–20% of cases, ECs may behave as a high-risk disease, with an aggressive clinical outcome. The first classification of EC, proposed by Bokhman, includes Type I (endometrioid-type) and Type II (serous-type) ECs [3]. In 2013, The Cancer Genome Atlas (TCGA) Research Network performed whole exome sequence analysis, transcriptome sequence analysis, genomic copy number analysis, protein array analysis, microsatellite stability testing and methylation profiling on 373 EC samples, including 307 endometrial endometrioid carcinoma (EECs), 53 serous endometrial carcinomas (SECs) and 13 mixed cases [4,5]. Based on their findings, four categories of ECs were identified: *POLE*/ultramutated, microsatellite instability (MSI)/hypermutated, copy-number low/NMSP and copy-number high/serous-like. To improve the clinical applicability of the TCGA classification, cheaper surrogates of molecular prognostic markers (in particular, the immunohistochemical assessment of MMR proteins and p53) have defined four groups, reflecting the TCGA prognostic groups: *POLE*-mutated (*POLEmut*), MMR-deficient (MMRd), p53-abnormal (p53abn) and no specific molecular profile (NSMP). These four molecular categories are essentially non-overlapping, although about 3% of cases appears to fall into multiple groups, showing two or more molecular signatures (multiple classifiers): in order of frequency, MMRd+p53abn, *POLEmut*+p53abn, MMRd+POLEmut, MMRd+*POLEmut*+p53abn. The clinical outcome is predicted by the driver molecular subtype; in detail, the *POLEmut* signature prevails over the other signatures, leading to a better prognosis regardless of MMR and p53 status; similarly, the MMRd signature supersedes p53 abnormalities. In this complex scenario, *POLEmut*, p53mut and MMRd can also be secondary molecular events. To date, molecular profiling is of particular importance and highly recommended in high-grade EECs and in intermediate–high risk ECs, in particular for therapeutical issues (possibility of de-escalation or to intensify treatments). However, a complete molecular classification surrogate (*POLEmut*, MMRd, NSMP, p53abn) is encouraged in all cases of EC for prognostic risk-group stratification, data collection, LS screening purposes, as potential influencing factors for adjuvant and systemic treatment decisions and for a predictive value for the ICI approach. According to de Biase et al. [6], an adequate diagnostic algorithm should include immunohistochemical evaluation of MMR proteins and p53 expression in all cases of EC, whereas *POLE* sequencing is to be restricted to early-stage cases with at least one of the following histopathologic features: (i) non-endometrioid histotypes (i.e., dedifferentiated/undifferentiated carcinoma), (ii) high grade, (iii) substantial LVSI, (iv) Stage IB-II. The reduction in tests would reduce the costs of molecular analysis, thus providing a better allocation of resources, without compromising the accuracy of risk grouping [7]. This algorithm may allow healthcare providers to follow the current ESGO-ESTRO-ESP guidelines [8], which recommend (i) avoiding adjuvant treatment for low- and intermediate-risk patients, including patients with high-grade and/or Stage II *POLE*-mutated EC; (ii) adding adjuvant brachytherapy or EBRT (external-beam radiation therapy) for high–intermediate-risk patients, especially in cases with significant LVSI and/or Stage II; (iii) reserving EBRT with concurrent adjuvant chemotherapy, or alternatively sequential chemotherapy and radiotherapy, for high-risk patients. Advanced-stage ECs are considered high risk regardless of molecular subgroups (including *POLE* tumors) and require adjuvant treatment. In fact, this algorithm does not recommend molecular analysis in advanced-stage cases, although several other studies suggest that molecular classification may guide the appropriate adjuvant treatment also in high-risk/advanced stages [9]. These unsolved issues will be better clarified by the Postoperative Radiation Therapy in Endometrial Carcinoma (PORTEC) trials [10,11,12], TAPER trial [13] and TransPORTEC RAINBO program [14]. The MSI/hypermutated group includes 30% of ECs and has a high mutational rate (18 × 10^−6^ mutations per megabase). This TGCA subgroup includes both low-grade and high-grade endometroid ECs, but also other histotypes, and its overall prognosis is generally intermediate [15,16]. To date, immunohistochemistry for MMR proteins (MLH1, MSH2, MSH6 and PMS2) represents the gold-standard surrogate to identify the MSI hypermutated group. Concerning the therapeutic option, based on their high mutational load and the rich immune infiltrate, MMRd ECs represent optimal candidates for immunotherapy. In this regard, MMRd solid tumors showed a significant response to immune checkpoint inhibitors such as Pembrolizumab and Dostarlimab [17,18,19,20]. Given the above-mentioned prognostic and therapeutic implications of MMR deficiency in EC, the aim of the present review is to provide an update on the current knowledge of MMR testing in EC. A systematic search of online databases, including EMBASE, MEDLINE and Cochrane, has been conducted. Articles published in English up to November 2023 were included. The following queries were used in the search: “endometrial cancer”, “mismatch repair deficiency”, “MMRd”, “microsatellite instability”, “MSI”, “immunohistochemistry”, “gene mutations”, “TCGA molecular groups”, “prognostic biomarkers”, “predictive biomarkers” and “artificial intelligence”. We also manually searched the eligible studies, and the included studies were managed using Zotero (ver. 6.0.30). Finally, we revised all the selected studies, in order to propose a practical guide for the pathological assessment and reporting of MMR testing in EC.

## 2. Molecular Landscape of MSI/MMRd EC

The MSI/hypermutated group, accounting for about 30% of ECs, is characterized by MSI, mostly caused by MLH1 promoter methylation, and a high mutational rate (18 × 10^−6^ mutations per megabase, with a high frequency of insertions and deletions), but low copy-number variations. Thus, MSI is defined as a condition of genetic hypermutability resulting from a defective DNA mismatch repair process, and the two terms are often used interchangeably [21]. MSI occurs when, during the DNA replication or in case of iatrogenic damage, frame-shift mutations (insertions or deletions) in MMR genes involve the short repetitive DNA sequences of 1–10 nucleotides (microsatellites or short tandem repeats), distributed along the genome of both coding and non-coding regions, being particularly sensitive to DNA mismatching errors, with a subsequent increased mutational burden and MMR deficiencies. MSI can be caused by somatic or germline alterations [22,23]. Somatic alterations, accounting for 85% of cases, include biallelic epigenetic MLH1 hypermethylation (in about 77% cases of sporadic endometrial cancers); downregulation of MMR genes by microRNAs; biallelic mutations; one somatic mutation and LOH; and secondary epigenetic MSH6 silencing induced by neoadjuvant RT/CHT. Germline mutations, accounting for 5% of cases, involve MMR genes and can determine two different types of clinical syndrome:-Constitutional mismatch repair deficiency (CMMRD) [24], a rare childhood cancer predisposition syndrome with recessive inheritance, due to a biallelic MMR gene mutation in which MMR defects (occurring in MLH1, PMS2, PMS1, MSH2 or MSH6) are inherited from both parents [25];-Lynch syndrome (LS), an autosomal dominant disorder characterized by the occurrence of multiple cancers, resulting from constitutional germline mutations, affecting the DNA MMR genes MLH1, MSH2, MSH6 and PMS2; constitutional MLH1 hypermethylation; or deletion of the stop codon (3′ end truncating) of the *EPCAM* gene causing the epigenetic silencing of the neighboring MSH2.

Generally, the MLH1 variant is correlated with the highest risk of colorectal cancer, while the MSH2 variant is correlated with the highest risk of other cancers [26]. ECs occurring in this setting represent 3–5% of cases and, even if they may occur at any age, they often arise in young women (45–55 years). EC is the index cancer in slightly more than 50% of cases. The most common target genes that harbor MSI in endometrial cancer include TP53, FBXW7, CTNNB1, ARID1A, PIK13CA, PIK3RI, PTEN, RPL22, PTEN, KRAS, ATR, CHK1, CDC5, Caspase 5, BAX gene, and JAK1 mutations. The lifetime risk of developing EC in LS patients is up to 71%; therefore, the detection of LS in patients affected by EC is crucial for genetic counseling and for the early diagnosis of secondary malignancies. According to the 2014 Clinical Practice Statement proposed by the Society of Gynecologic Oncologists, systematic clinical screening, including personal and family history and molecular/IHC screening, should be performed in all women diagnosed with EC [27,28].

The identification of MMR pathogenic variants in germline sequencing is the gold standard for the diagnosis of LS. However, the first step for LS screening in EC is represented by immunohistochemistry. In this regard, most of the ECs with MSI/MMR deficiency shows MLH1 and PMS2 loss related to sporadic *MLH1* promoter hypermethylation (met-ECs) [29,30]; the remaining MMRd EC cases may be related to LS (mut-EC9). If the loss of MLH1 is detected by immunohistochemistry, testing for the presence of *MLH1* promoter hypermethylation should be performed in order to detect sporadic *MLH1* loss unrelated to LS [31]. Moreover, interpretation of the loss of MLH1 expression should be performed with caution:-Homozygous MLH1 promoter hypermethylation is predominantly associated with sporadic cases;-Heterozygous signature of the MLH1 promoter hypermethylation, as a second-hit event results in the loss of expression of the wild-type allele in LS tumors;-The MLH1 pathogenic variant can be associated with MLH1 promoter hypermethylation.

The presence of MLH1 promoter hypermethylation should not rule out de facto the possibility of an LS diagnosis. MLH1 promoter methylation is known to be an aging-related event, thus for early-onset cancer or in case of familial history of EC, molecular testing should be performed regardless of the MLH1 promoter hypermethylation. In cases where sporadic MLH1 hypermethylation is excluded, patients are then referred for genetic counseling and germline genetic testing to confirm the diagnosis of LS. Finally, in the absence of a germline mutation, somatic mutations have also been investigated.

The two main forms of MSI (mutational and epigenetic) identify different clinical characteristics of ECs, offering a possible substratification of the MMRd EC group. “Met-EC” are generally constituted by a lower proportion of Grade 1 (37.5%) and a higher proportion of Stage III/IV tumors (37.2%). Their overall and progression-free survival are significantly worse than those of “suspected-LS” (cases with other MMR protein loss and MLH1/PMS2 loss without MLH1-silencing) [32,33]; in this way, “met-ECs” would be the main target for anti-PD-1 antibody treatment. On the other hand, patients with mut-ECs are associated with a higher trend towards a higher mutation burden, better recurrence-free survival, longer survival, higher response rates and higher risk for second cancers compared with patients with met-EC. Longitudinal single-cell RNA-seq of circulating immune cells revealed contrasting modes of anti-tumor immunity and tried to explain how the interplay between tumor-intrinsic and extrinsic factors influence the ICB response. Effector CD8+ T cells were correlated with the regression of mutational MMRd, while activated CD16+ NK cells were associated with ICB-responsive epigenetic MMRd tumors [34].

Finally, in up to 59% of patients displaying MSI and/or MMRd (‘suspected LS’—sLS), germline variants affecting function or promoter hypermethylation of the *MLH1* gene cannot be detected. In tumors with unexplained MMRd/MSI/MLH1-unmethylated tumors, *POLE*/*POLD1* germline and somatic screening may serve as a marker for the sporadic origin of the disease. It is important to recognize that the presence of *POLE* EDM may be a novel alternative pathway of MSI in ECs, generally somatic, but it does not exclude the possibility of germline MMR mutation.

## 3. The Histo-Molecular Approach

### 3.1. MMR Deficiency as a Predictive and Prognostic Biomarker in Endometrial Cancer: The Relationship with Molecular and Histological Subtypes

Endometrial cancer management has greatly benefited from histopathological classification, based on the histological subtype and tumor grade of differentiation, which has allowed for prognostic stratification into discrete risk categories and guided adjuvant and surgical therapy. Low-grade (G1–G2) endometrioid endometrial carcinomas (EEC) represent the subset with the most favorable outcome. High-grade (G3) endometrioid endometrial carcinoma has demonstrated an intermediate prognosis. All the other non-endometrioid subtypes are considered as high-grade (G3) and display an aggressive behavior. This group includes many histological subtypes, some long-known, such as serous endometrial carcinomas (SEC), clear cell endometrial carcinoma (CCEC) and mixed endometrial carcinoma (MEC), but also others recently classified as undifferentiated/dedifferentiated endometrial carcinoma (UEC/DEC) and uterine carcinosarcoma (UCS). This group also includes rarer subtypes, such as neuroendocrine endometrial carcinoma (NEEC), mesonephric-like endometrial carcinoma (MLEC), and gastric/gastrointestinal-type endometrial carcinoma (GTEC) [35]. Additional significant histopathological prognostic markers have been utilized to adjust for risk, particularly in endometrial cancer (EEC), such as myometrial infiltration and lymphovascular space invasion (LVSI) [8]. Unfortunately, the pathologic evaluation alone, even though playing a fundamental role in prognostic stratification, has shown some limits, such as the imperfect reproducibility of grading determination, frequent histological overlapping between subtypes and suboptimal interobserver agreement (particularly among high-grade subtypes) [36]. The TCGA classification in four molecular groups (*POLE*/ultramutated; MSI/hypermutated; copy-number low/endometrioid; copy-number high/serous-like) provided novel revolutionary insights into risk stratification, discovering new predictive and prognostic biomarkers and allowing a more precise characterization of patients’ outcomes [4]. The *POLE*/ultramutated group is defined by somatic mutations in the exonuclease domain of DNA polymerase epsilon (*POLE*) and is characterized by a very high mutation rate (232 × 10^−6^ mutations per megabase). This group includes both low-grade and high-grade EECs, all showing an excellent prognosis and no recurrence, independent from the FIGO grade. The MSI/hypermutated group is defined by microsatellite instability and shows a high mutational rate (18 × 10^−6^ mutations per megabase). Similarly, this group includes both low-grade and high-grade EECs, comprehensively presenting an intermediate prognosis. The copy-number low/endometrioid group presents no specific mutations, being characterized by the absence of *POLE*, MMR and TP53 mutations and a low degree of somatic copy-number alterations (SCNA). This group mainly includes EECs and shows an intermediate overall prognosis. The copy-number high/serous-like group is characterized by TP53 mutations (90% of cases) and a high SCNA. This group mainly includes SECs and shows a poor overall prognosis [4]. As demonstrated by subsequent studies, the TCGA classification may be predicted using cheaper immunohistochemical surrogates of molecular prognostic and predictive markers. In fact, the immunohistochemical assessment of p53 and MMR protein expression is used as a surrogate for the identification of the copy-number-high/serous-like group and MSI/hypermutated groups, respectively. Unfortunately, a reliable surrogate of *POLE* sequencing has not yet been identified. However, the surrogate classification defines four groups reflecting the TCGA molecular groups: *POLE*-mutated (*POLEmut*, surrogate of *POLE*/hypermutated), MMR-deficient (MMRd, surrogate of MSI/hypermutated), no specific molecular profile (NSMP, surrogate of copy-number low/endometrioid) and p53-abnormal (p53abn, surrogate of copy-number high/serous-like). According to the TCGA classification, the main histological subtypes of EC are distributed as follows: low-grade EECs (6% *POLEmut*, 25% MMRd, 64% NSMP, 5% p53abn); high-grade EECs (12% *POLEmut*, 39% MMRd, 28% NSMP, 21% p53abn); SECs (100% p53abn); CCECs (4% *POLEmut*, 10% MMRd, 42% NSMP, 44% p53abn); UECs/DECs (12% *POLEmut*, 44% MMRd, 25% NSMP, 19% p53abn); and UCSs (5% *POLEmut*, 7% MMRd, 14% NSMP, 74% p53abn) [9,10,37,38].

MMRd/MSI EC accounts for about 25–30% of ECs [39], showing distinctive histopathological features such as (i) lower uterine segment origin; (ii) endometrioid differentiation; (iii) severe nuclear atypia with undifferentiated component; (iv) high mitotic index; (v) high tumor-infiltrating lymphocytes (TILs) and/or peri-tumoral lymphocytes: ≥40 TIL/10HPFs, with more CD8+, CD45RO+ and PD1+ T cells at the invasive tumoral margin in mut-ECs compared with met-ECs; (vi) high morphological heterogeneity; (vii) substantial lympho-vascular space invasion (LVSI); (viii) deeper myometrial invasion; and (ix) synchronous ovarian cancer (clear cell or endometrioid variants) (Table 1).

Regarding the prevalence of MMRd ECs across the different histotypes of EC, undifferentiated/dedifferentiated carcinoma (UEC/DEC) is the most common MMR deficient subtype (44%) followed by neuroendocrine carcinoma (42.9%), high-grade endometrioid carcinoma (39.7%), mixed forms (33.3%), low-grade endometrioid carcinoma (24.7%), clear cell carcinoma [40] (9.8%) and carcinosarcoma [41] (7.3%). Only sporadic cases of serous carcinoma and mesonephric-like carcinomas have been reported to show MMR deficiency (Table 2).

Compared to *POLEmut* ECs, MMRd ECs seem to be more prognostically affected by clinicopathological variables, although not as much as NSMP ECs. The ESGO-ESTRO-ESP guidelines [8] substratify MMRd ECs into different risk groups based on pathological features, such as the depth of myometrial invasion, LVSI and histotype. On the other hand, in the MMRd molecular group, grading does not matter. The overall prognosis of MMRd ECs is intermediate across different histotypes leading to worsened outcomes (with higher risk for relapse) in early-stage, low-grade EECs, intermediate prognosis in high-grade EECs and improved outcomes in non-endometrioid carcinomas (NECs) [42].

Interestingly, a recent study focused on EECs characterized by a distinctive myometrial pattern of invasion, namely the microcystic elongated, angulated and fragmented (MELF) pattern, has shown a higher frequency of MSH2-MSH6 loss in this group (7.14% in MELF+ vs. 3.96% of MELF-), suggesting a possible different molecular signature among cases with and without the MELF pattern of invasion. Moreover, as described in this study, MMR deficiency could affect the risk of nodal metastases for tumors of the same size in the MELF- population, but not in MELF+ ECs [43].

Considering that the rare MMRd serous ECs or MMRd ECs with serous features have a prognosis comparable to MMRd EECs, a similar management seems to be necessary. Different studies [40,41] describe different percentages of MMRd CCECs, but the MMRd signature has been more frequently described in mixed EEC and CCEC [42]. Although the ESGO/ESTRO/ESP guidelines include CCEC among the non-endometrioid subtypes, the p53abn CCEC is characterized by a poor outcome, while the MMRd and NSMP outcome still needs to be more clearly defined. Regarding UEC/DECs and UCSs (in particular those ones with a UEC/DEC component), they may sometimes display a better prognosis [44]. As regards NEECs, MMRd represents the most common signature and, interestingly, the more frequent MMRd mixed EEC/NEECs seem to be prognostically similar to their EEC counterpart [45]. On the contrary, up until now, MLEC appears to not have had an MMRd signature [46]. As regards the gastro-intestinal differentiation in EECs, according to some studies they might be associated with an MMRd signature, having a poorer prognosis [47]. Instead, less is currently known as regards the prognosis and MMRd signature in the pure gastro-intestinal type of EC (GTEC).

Synthetically, the MMRd group prognosis seems to be intermediate across different histological EC subtypes. In EECs usually having a good prognosis (early-stage, low-grade), MMRd represents a risk factor for recurrence [48]. Conversely, in high-grade EECs, MMRd is associated with an intermediate prognosis [49,50]. In non-endometrioid carcinomas, which typically are considered aggressive, MMRd is a favorable prognostic factor [51,52,53,54]. The current evidence suggests that the MMRd group may be considered as an intermediate-risk group regardless of the histological subtype. An exception would be UEC/DEC, in which a loss of SWI/SNF protein expression appears to be associated with aggressive behavior even in the case of an MMRd signature [7,55].

### 3.2. EC Histological Subtypes and Genomic Alterations: The Relationship with Molecular Classification and MMR Deficiency

In the past decade, targeted gene and exome sequencing have also allowed researchers to uncover additional genetic alterations, specifically correlated with each histological subtype and TCGA subgroup [56]. Overall, EECs are characterized by frequent alterations of the PI3K–PTEN–AKT–mTOR, RAS–MEK–ERK and WNT–β-catenin pathways. Moreover, the *ARID1A* tumor suppressor gene is also frequently dysregulated [57]. In a recent study by Da Cruz Paula et al., *PTEN* (86%), *ARID1A* (66%), *PIK3CA* (56%), *PIK3R1* (34%) and *CTNNB1* (27%) were found to be the most commonly mutated genes in the endometrioid histological subtype [58]. A step-wise increment in the frequency of specific driver mutations was observed in FIGO Grade 1, Grade 2 and Grade 3 EECs, including *ARID1A* (54%, 80% and 90%, respectively), *KMT2D* (14%, 26% and 80%, respectively) and *TP53* (8%, 14% and 50%, respectively) [58]. As previously discussed, a relatively high incidence of *POLE* mutations and a high rate of MSI, reflecting MMR protein defects, are detectable in EECs. Mutational signature analysis in EECs revealed that 80% of *POLEmut* cases had a dominant signature associated with *POLE*, while the other 20% had dominant signatures associated with aging or MMRd. Of the MMRd EECs, 68% had a dominant signature associated with MMR deficiency, whereas the remaining 32% showed a dominant signature associated with aging [58]. In this study, several differences in mutational profiles between early- and advanced-stage EECs were identified. Early-stage EECs were more likely to harbor *POLE* mutations and *POLE* signatures, but showed a lower incidence of MMRd-related mutational signatures. Moreover, early-stage EECs had a higher frequency of *PTEN* mutations. Conversely, advanced-stage EECs more frequently presented *JAK1*, *ARID1B*, *SOX17* and *MDC1* mutations. After excluding MSI-high and *POLEmut* cases, an even higher incidence of *SOX17* alterations has been found in advanced-stage EECs [58]. In sporadic EECs, MSI is mainly due to *MLH1* gene epigenetic silencing as a consequence of promoter hypermethylation. This alteration results in MMR deficiency and the accumulation of somatic mutations throughout the genome. Some of these mutations may represent pathogenic driver events. Recent studies have described *ATR*, *CTCF*, *JAK1*, *RNF43* and *RPL22* as driver genes that are frequently mutated in MMRd EECs [59,60,61,62,63]. Mutations in the *TP53* gene are the most frequent molecular aberrations in serous carcinomas, occurring in >85% of the cases and representing an early pathogenetic event in this histological subtype [64,65,66]. In addition to *TP53* mutations, other somatic mutations in SECs involve the *PPP2R1A*, *FBXW7*, *SPOP*, *CHD4* and *TAF1* genes; *ERBB2*, *MYC* and *CCNE1* amplifications and p16 and synuclein-γ overexpression have also been described. The druggable PI3K pathway may also be altered in SECs, more frequently because of *PIK3CA* mutations, less frequently due to *PTEN* or *PIK3R1* mutations [56]. In the study by Da Cruz Paula et al., *TP53* (94%), *PPP2R1A* (41%), *PIK3CA* (35%) and *FBXW7* (18%) were the most frequently mutated genes in SECs. *ERRB2* alterations (hotspot mutations and amplification) and *CCNE1* amplification were observed in 29% and 18% of SECs, respectively. In this study, no differences in mutations and copy-number alterations have been found between early-stage and advanced-stage SECs. However, a numerically higher frequency of *ERBB2* amplification were observed in advanced-stage SECs [58]. CCECs were not included in the histological subtypes analyzed by TCGA; therefore, the molecular features of this subtype remain less studied in comparison with EECs or SECs. However, in the studies currently reported in the literature, *TP53* has been found mutated in 31–50% of cases. MSI and abnormal MMR protein expression have been detected in 0–19% of cases. Other described mutations regard *PPP2R1A* (16–32%), *PIK3CA* (14–37%), *FBXW7* (7–27%), *PTEN* (0–25%), *KRAS* (0–13%), *ARID1A* (14–22%), *SPOP* (14–29%) and *POLE* (0–6%). Additionally, genomic gains have been described for *CCNE1* (18%), *ERBB2* (11%) and *CEBP1* (11%), whereas deletions have been reported for *DAXX* (11%) [52,67,68,69,70,71]. As regards UCSs, *TP53* represents the most commonly mutated gene (64–91%). Other frequent mutations regard *FBXW7* (11–38%), *PTEN* (18–47%), *PIK3CA* (15–41%), *CHD4* (16–17%), *ARID1A* (10–24%), *KRAS* (9–29%), *PPP2R1A* (13–27%) and *FOXA2* (5–15%). Other genes that are putative drivers of uterine carcinosarcoma are RB1 (4–11%), U2AF1 (4%), ZBTB7B (11%), ARHGAP35 (11%), SPOP (7–18%), HIST1H2BJ (7%) and HIST1H2BG (7%). Interestingly, RB1, U2AF1, and ZBTB7B are considered to be driver genes in UCSs but not in SECs or EECs. Moreover, a copy-number gain on chromosome 5p, including the TERT cancer gene, is more frequently present in UCSs compared to other histological subtypes (50% versus 17%, respectively). *POLE* mutations have been found in only 2–4% of UCSs. MSI has been observed in a variable percentage of cases (3.5–21%) [53,54,72,73,74,75,76,77]. A recent study by Asami et al. analyzed 1029 patients with endometrial cancer, investigating different genetic alterations in the four molecular subtypes and correlating them with prognosis [78]. *TP53* mutations were significantly more common in the p53abn group than in the other three groups. *PTEN* and *ARID1A* mutations were significantly less common in the p53abn group compared to the other groups. *KRAS* mutations were found more frequently in the NSMP group. No gene mutations were found to be more frequently associated with the MMRd group [78].

### 3.3. MMR Deficiency in Light of 2021 ESGO-ESTRO-ESP Guidelines and 2023 FIGO Staging System: A Combined Histo-Molecular Approach for Risk Stratification

Figure 1 shows a diagrammatic representation and an algorithmic approach of how MMRd and the other molecular groups may influence the outcome, when combined with histological subtype and clinicopathological variables, according to the ESGO-ESTRO-ESP risk groups.

The updated 2023 FIGO staging of EC combined molecular classification and the various histological types to better reflect the complex nature of endometrial carcinomas and their biological behavior [79]. Together with molecular classification, and perhaps even more importantly, histopathological features play the central role in the 2023 FIGO staging of EC. Histological subtype is an important prognostic factor. In this revised FIGO staging, histological subtypes are divided into non-aggressive (i.e., low-grade EECs), and aggressive histological types (i.e., high-grade EECs, SECs, CCECs, MECs, UECs/DECs, UCSs, mesonephric-like and gastrointestinal-type mucinous carcinomas). Notably, high-grade EEC is a prognostically, clinically and molecularly heterogenous category, and hence is the subtype which benefits the most from molecular profiling. Otherwise, without molecular profiling, high-grade EECs cannot be included into a specific risk group. Specifically, *POLEmut* high-grade EECs show an excellent prognosis, and p53abn high-grade EECs have a bad prognosis. Conversely, it has been demonstrated that, irrespective of grading, the MMRd group have an intermediate prognosis, whereas NSMP high-grade EECs, particularly if estrogen receptor-negative, always display a bad prognosis [80,81]. The new 2023 FIGO staging system for EC, based on combined molecular and histological findings, is summarized in Table 3.

According to the 2023 FIGO staging of EC, MMRd, similar to NSMP status, does not modify the early FIGO stages (I and II). Instead, the presence of *POLE* mutations or *TP53* alterations now modifies the FIGO stage. As regards Stage I and II tumors, *POLEmut* ECs are now classified as Stage IAm_POLEmut_, independent from LVSI or histological subtype. Instead, as regards p53abn tumors with the same features, they are directly upstaged and classified as Stage IICm_p53abn_. In the case of multiple classifiers with *POLEmut* or MMRd and secondary p53 abnormality, tumors should be considered as *POLEmut* or MMRd, and staged accordingly. As regards advanced FIGO stages (III and IV), the staging is not altered by molecular characterization. However, Stage III and IV tumors belonging to the p53abn group should be reported as Stage IIIm_p53abn_ or Stage IVm_p53abn_, respectively, for data collection purposes. Additionally, the same has to be conducted for MMRd tumors, which should be recorded as Stage IIIm_MMRd_ or Stage IVm_MMRd_ for data collection and, more importantly, in view of its predictive value for treatment with immune checkpoint inhibitors and the demonstrated progression-free and (preliminary) overall survival benefit [79].

## 4. Immunotherapy for MSI/dMMR Gynecological Cancers

Before the “immunotherapy era”, advanced-stage and recurrent/metastatic ECs have shown a limited response to cytoreductive surgery and systemic therapy. However, in the last few years, several studies have demonstrated that MSI/MMRd ECs are unlikely to respond to conservative hormonal treatment, show a high likelihood of LVSI justifying a sentinel or other nodal procedure and have a good response to RT (including just VBT in the absence of unfavorable risk factors). The efficacy of immune checkpoint inhibitors (ICI) especially in endometrial tumors showing high mutational burdens and immune cell infiltration (immunologically ‘hot’ tumors) has also been documented. In this regard, the MSI/hypermutated group represents the best candidate for immunotherapy.

In detail, when MMR proteins are deficient, the accumulation of uncorrected DNA mutations determines the expression of novel neoantigens and a high tumor mutational burden; these events produce an increased inflammatory response.

Based on these findings, several studies have reported the results of immune checkpoint inhibitors (anti-PD-L1 antibody) in EC with MSI. In detail, the Keynote-158 study [17] demonstrated the antitumor activity and improved survival of pembrolizumab with manageable toxicity in patients with previously treated, advanced MMRd/MSI-H ECs. Therefore, pembrolizumab received FDA approval for advanced ECs showing disease progression despite systemic therapy in any setting and which are not candidates for surgery or radiation. Following the significant clinical benefits demonstrated in the RUBY/ENGOT-en6/GOG-3031/NSGO Phase III trial in patients with advanced or recurrent ECs (72% and 36% decrease in the disease progression and death risk, in dMMR/MSI-H ECs and in the overall population, respectively), the anti-PDL-1 dostarlimab has been approved by the FDA for advanced MMRd ECs using a specific companion diagnostic assay (Ventana MMR Dx). The two randomized Phase III trials (ENGOT-en6/GOG-3031/RUBY and NRG-GY018/Keynote-868) have demonstrated a statistically significant and unprecedented PFS advantage with the addition of an immune checkpoint inhibitor (ICI) (dostarlimab or pembolizumab, respectively) to standard carboplatin/paclitaxel chemotherapy followed by ICI maintenance therapy in MMRd patients with a hazard ratio (HR) of 0.28 (95% confidence interval [CI] 0.16–0.5) and 0.30 (95% CI 0.19–0.48), respectively. Positive results have also been documented with other anti-PD-L1 therapies including nivolumab, atezolizumab, avelumab and durvalumab.

Moreover, several ongoing studies are investigating the efficacy of pembrolizumab and dostarlimab in advanced/recurrent MMRd/MSI-H ECs which have not previously received first-line chemotherapy. Improved ORR, PFS and OS have been observed in Lynch-associated vs. sporadic MMRd ECs when treated with pembrolizumab. A correlation between MLH1-hypermethylation and poor response to single-agent immunotherapy with pembrolizumab has also been described. Finally, according to the GARNET trial [82], MSI-H tumors regardless of mutation or presumed methylation had a similar efficacy with single-agent immunotherapy with dostarlimab.

## 5. MMRd/MSI ECs: Testing Method

The terms “MMR deficiency” and “MSI” have been used interchangeably, even if they refer to results obtained from different assays which have variable performance characteristics in different tumor types. MMR deficiency is conventionally defined by the immunohistochemical loss of expression of one or more of the four main MMR proteins (MLH1, PMS2, MSH2 and MSH6), with clear evidence of an internal positive control. These proteins form two stable heterodimers, which are MLH1-PMS2 and MSH2-MSH6. As a general rule, MLH1 loss is typically accompanied by PMS2 loss; on the other hand, MSH2 loss is associated with MSH6 loss. The PMS2 antibody detects all cases that harbor either MLH1 or PMS2 abnormalities; the MSH6 antibody detects all cases that harbor either MSH2 or MSH6 abnormalities. MLH1 and MSH2 alone do not recognize cases with PMS2 or MSH6 abnormalities, because PMS2 and MSH6 can be absent without affecting the expression of the other protein of the pair (in particular, PMS2 can be replaced by PMS1 or MLH3; MSH6 can be substituted by MSH3). Although different pathological studies and meta-analyses have demonstrated that a combination of MSH6 and PMS2 antibody for immunohistochemical use may allow a reduction in the cost without a decrease in the diagnostic accuracy, the four-antibodies platform is considered the best way to direct genetic testing [83,84]. MMR protein immunohistochemical analysis remains the gold standard for EC molecular assessment because of many advantages including IHC’s wide availability in almost all laboratories and the lower cost compared to NGS or PCR and high concordance (94%) with molecular analysis. Moreover, literature data have pointed out that MSI testing has a decreased sensitivity for MSH6 mutations which are frequently encountered in ECs. However, there are specific conditions requiring PCR or NGS instead of MMR protein immunohistochemistry, namely:-Whenever IHC shows indetermined/ambiguous/equivocal results;-False negative IHC results due to pre-analytical tissue poor fixation;-Whenever IHC shows aberrant patterns (e.g., cytoplasmic, dot-like and perinuclear staining);-Whenever IHC shows the loss of only one heterodimer subunit (i.e., only MLH1 or PMS2 and not both).

## 6. Practical Issues in IHC MMR Proteins Interpretation: When–Where

As previously mentioned, the first method for MSI/MMRd testing is represented by IHC, largely supported by the availability of antibodies in almost all laboratories (Figure 2). However, molecular testing should be performed in all cases of indeterminate IHC results, together with germline testing in the presence of a strong family history.

When and where: IHC can be performed on both biopsies and surgical specimens. The final aim is always the same: the choice of the best-preserved sample including an internal positive control [85]. The advantages of choosing biopsies are the better degree of fixation of tissues and the early knowledge of MSI status in a pre-operative setting. The advantages of choosing surgical specimens are the larger amount of tumoral tissue and the possibility to overcome tumor heterogeneity. Indeed, morphological heterogeneous tumors are often molecularly heterogeneous, with a predominant p53-mutated clone in the solid component and a p53 wild-type cell population in the glandular area.

MMR IHC (typical combinations): any profile besides the classical typical MMRd phenotype (Table 4) was defined as unusual. In total, 15% of MMRd tumors present with an unusual phenotype [86], specifically:(a)Isolated loss of PMS2 or MSH6 regardless of the microsatellite status;(b)Classical loss of MLH1/PMS2 or MSH2/MSH6 without MSI (MSS tumors) or with MSI low (false positive cases: MSS/MSI-L-MMRd, due to MLH1 promoter hypermethylation or somatic MMR variants);(c)Four MMR proteins retained expression (non-functional protein with retained antigenicity) with MSI (false negative cases: MSI-MMRp in the case of *POLE* variants or missense mutation of MMR proteins, in particular MLH1);(d)Complex loss of MMR proteins regardless of the microsatellite status.

Some cases may rarely show a loss of two discordant MMR proteins, different from the usual couples represented by MLH1-PMS2/MSH2-MSH6. In other cases, a loss of three or all four MMR proteins may also be observed. These cases may be the consequence of different possible mechanisms. One of the most frequent events is a super-added MLH1 promoter methylation occurring as a sporadic alteration in any MMRd, independent from the cause. Alternatively, unusual combinations of MMR loss may be observed in *POLEmut* ECs, where multiple MMR genes mutations may occur as secondary events and may show a subclonal pattern of expression, as previously explained. Another possible mechanism underlying unusual combinations is represented by the presence of other somatic mutations. In some cases of LS-ECs with known germline MSH6 mutations, it is sometimes possible to observe a combined MSH6 and MSH2 loss of immunohistochemical expression, which is thought not to be related to MSH2 mutations. It has been postulated that somatic MSH3 mutations, and the subsequent lack of an MSH2 stabilizing partner, might be responsible for unexpected MSH2 loss in such cases [87].

A recent study has shown that the proportion of non-CRC was higher in the unusual (32.6%) than in the classical MMRd group (13.1%) and that genetic syndromes were significantly more frequent in unusual (44.9%) than in classical MMRd patients (21.4%). The main genetic syndrome was LS, but two other syndromes were identified in the unusual MMR-D group: *POLE* deficiency and CMMR-D (constitutional MMR deficiency). In particular, the ECs show with high-frequency phenotypes a and b. In this way, classification of unusual MMRd patterns helps to identify not only MMR deficiency but also a high frequency of genetic syndromes who could benefit from ICI.

## 7. Problems of IHC Interpretation

### 7.1. Poor Fixation, Cauterization Artefacts, Neoadjuvant Chemotherapy and Freezing of Tissues

They affect IHC detection and diagnostic outcome of MSI testing in EC, generally showing a gradual decrease in intensity from positive areas (Figure 3). Staining protocol should be standardized with appropriate quality controls.

#### Solutions

Search for well-preserved tissue: Well-fixed areas should be examined when reporting MMR IHC, to avoid erroneous interpretation; moreover, we have to remember that areas of cauterization could show increased cytoplasmic staining and a loss of nuclear staining, but also preservation of nuclear staining, making interpretation difficult. Such areas such should be interpreted with caution and other areas of the tumor should be used to assess.

Look at the surrounding normal background: MMRd non-neoplastic endometrial glands were identified in 47–70–97% of LS patients and in no patients without LS (*p* < 0.001), generally seen as large contiguous groups and at a higher density. In patients with LS, MMRd non-neoplastic endometrial glands may accumulate immunogenic frameshift neoantigens resulting from mutations (ins/del) at coding microsatellites. The body’s immune response to these frameshift neoantigens may result in the eradication of the MMRd glands before stepwise progression to neoplasia. Immunogenic frameshift neoantigens are a potential target in developing vaccines for the chemoprevention of MSI-H/LS-related cancers, including EC. The evaluation of MMR protein expression of benign background endometrium in EC patients may be further explored as a possible useful addition to the Lynch syndrome screening algorithm [88].

Look at the surrounding precancerous background: AEH/EIN associated with MMR loss and *POLEmut* is rarely diagnosed except when adjacent to carcinoma suggesting that AEH/EINs associated with these molecular subtypes, with their resulting high mutation rate, progress rapidly. Different studies have confirmed that MMRd EAH/EINs and MMRd early ECs are less responsive to conservative treatments, with a lower incidence of resolution and a higher incidence of progression and recurrence with progestin therapy. On this subject, only MMRd ECs with ‘low-risk’ features (e.g., progesterone and estrogen receptor-positive, no CNND1 mutation, no LS-related, no PTEN methylation) as well as *POLEmut* and a subset of p53 wild-type tumors may have improved response rates to a conservative approach, although this still needs to be validated.

### 7.2. Weak and or Focal Expression

Some missense mutations can result in weak/focal expression. Therefore, very weak staining/very focal expression (compared to internal control) (<5%; <10%) is best regarded as ‘loss’. An example can be represented by the weak focal/patchy immunoreactivity for MSH6 in the case of MSH2 germline mutations.

### 7.3. Numerous Lymphocytes and Stromal Cells

MMRd ECs typically have a large number of TILs. This high amount of tumor-associated inflammatory cells, along with endometrial stromal cells, should be distinguished from neoplastic cells, in order to avoid misinterpreting them as positive tumor cells. Fortunately, morphological criteria (cell dimensions and shape) are usually sufficient to differentiate inflammatory or stromal cells from tumor cells.

### 7.4. Internal Control

A fundamental point is the presence of intact internal control staining. Indeed, it is essential to compare normal and tumoral tissue.

*Normal results* refer to:-An internal control tissue staining being diffuse and at least faint and adjacent tumor cells with similar or stronger staining intensity throughout the tumors.

*Abnormal results* include those conditions in which:-There is a good and strong internal control but tumoral tissue stains less than 5%;-More than 10% of tumoral tissue presents a sharply demarcated lower staining intensity or absence of the staining compared with the evident internal control.

*Equivocal results* which require MSI analysis include cases showing:-Tumoral area with weaker staining than internal control tissue;-Very faint staining in both internal control and tumor tissue;-Any other ambiguous pattern after excluding fixation issues.

### 7.5. Heterogeneous Loss (Subclonal Pattern)

Some cases may show a heterogeneous loss of MMR protein expression, only limited to some tumoral areas. This particular pattern of expression has been defined as a “subclonal pattern” [89]. Stelloo et al. suggested labeling these tumors as “MSI” if the subclonal loss of expression involves at least 10% of the whole tumoral area. However, before labeling these tumors as MMRd tumors, it is mandatory to exclude fixation artifacts, always keeping in mind to have a look at the internal positive control. This rare kind of pattern is a result of tumor heterogeneity. Indeed, when different subclones emerge, the ones with a significant survival advantage are favored over other subclones and then gradually propagated.

Most commonly, this heterogeneous, geographical loss regards MLH1 and results from sporadic somatic (heterogeneous) methylation of its promoter (Figure 4).

In rare cases, also germinal MLH1 promoter methylation is observable, but these cases generally show a uniform loss of nuclear staining. As usual, PMS2 loss is associated with MLH1 loss and also generally shows a heterogeneous pattern, but, in some cases, PMS2 loss may be complete or may also present as isolated PMS2 loss.

The heterogeneous loss of expression may also regard MSH6. In fact, the MSH6 gene has a mutation-prone microsatellite, because Exon 5 shows a polycytosine tract (C8) which may easily undergo frameshift mutation in cases with MSI/MMRd [90]. This may lead to a so-called secondary “passenger mutation” in the *MSH6* gene, with a consequent subclonal pattern of MSH6 loss in MSI/MMRd cases due to any possible cause, i.e., sporadic mutation in an MSH6 coding mononucleotide tract, germline or *POLE* mutations [91].

Hence, pathologists may encounter two different situations, both characterized by a subclonal MSH6 loss. The first one is represented by a subclonal MSH6 loss associated with another MMR defect. In these cases, the reporting terminology should be as for the underlying defect, which drives the secondary subclonal MSH6 loss. The second situation is represented by an isolated subclonal MSH6 loss. These cases should always be reported as abnormal and may indicate an underlying germline alteration, more probably in a gene other than *MSH6*.

Therefore, in cases showing heterogeneous loss of MMR proteins, it is important to remember that the current suggested cutoff is 10% and that some cases may also have underlying germline defects. These cases should be reported as MMRd with subclonal MMR loss, even though the biological aspects (clinical behavior, response to therapy include immune modulation) are currently far from being completely understood.

### 7.6. Unusual Stainings

In some cases, it is possible to observe a cytoplasmic pattern of staining for MMR proteins. This pattern, although rare, is not to be considered as retained expression and represents an artifactual phenomenon. Similar considerations must be made for cases with nuclear granular dot-like staining. This pattern has been currently reported only with the M1-clone of the anti-MLH1 antibody (Roche Diagnostic) [92] and should not be interpreted as a positive result (Figure 5). Awareness of this clone-dependent artifact, even if rare, is important to prevent reporting mistakes and unnecessary referrals for germline mutation testing.

### 7.7. Unusual Combinations

See the above-mentioned atypical phenotypes.

### 7.8. Interlaboratory Reproducibility

As reported in the literature, there is a high level of interlaboratory agreement in molecular classification and MMR proteins immunohistochemical evaluation [93]. The most common reason for disagreement is attributable to possible variabilities in p53 staining. Similarly, in a less frequent proportion of cases, it may be attributable to interpretative errors in PMS2 or MSH6 staining. Even though interlaboratory agreement is usually high, several solutions may be proposed to further improve concordance. Among them are education practical courses, central quality control to assess technical quality and ensure immunohistochemical staining consistency and the use of secondary molecular testing in doubtful cases.

### 7.9. Interobserver Reproducibility

Interobserver agreement is reported to be very high in the literature [94]. Sari et al. reported a 96% agreement for MMRp cases and 82% agreement for MMRd cases [94]. Discrepant evaluations were observed in 5.4% of MSH6, 4% of PMS2, 4% of MLH1 and 1.3% of MSH2 stains. Discordant interpretations/troubling cases were associated with specific issues, represented by a heterogeneous subclonal staining pattern, high amount of intratumoral inflammatory cells, reduced staining in tumor and in internal control, scattered absent/weak staining adjacent to tumor cells with strong nuclear staining and variable intensity patterns throughout the tumor, cytoplasmic or other artefactual staining. Therefore, as already said before, it is important to be aware of these possible issues and to always consider them during immunohistochemical evaluation, to reduce mistakes and further increase interobserver agreement.

### 7.10. Primary Vs. Metastasis

An important point to be considered regards the eventual usefulness of re-testing for MMR proteins in cases of recurrent EC. In a study by Ta et al. [95] on 29 patients with metastatic EC, 14 cases (48.2%) were found to be MMRd at the metastatic sites. Furthermore, 2 cases (6.9%) showed a discordant MMR status, with PMS2 loss only present at the metastatic sites. In both of these cases, at the primary uterine site there was an abnormal subclonal loss, associated with MLH1 promoter methylation. Another study by Spinosa et al. [96], conducted on 43 patients, reported 3 cases (12%) showing MMRp status at the primary uterine tumor, who developed MMRd in the recurrent setting. On the basis of these studies, we may say that advanced endometrial cancer can rarely show the somatic loss of MMR protein expression in recurrent sites, compared to a matched paired primary tumor. This clonal evolution may have important therapeutic implications for patients with recurrent uterine cancer. For these reasons, it is recommendable to always re-evaluate the immunohistochemical expression of MMR proteins in cases of recurrent uterine cancer, to also identify the rare cases with discordant MMR expression at recurrence.

## 8. MMRd and Artificial Intelligence: Futuristic Approaches for MSI Detection

The assessment of EC histological subtypes, molecular groups and mutation status is fundamental for the diagnostic process, which consequently influences prognosis and treatment. The application and the ongoing development of artificial intelligence may provide a great help for pathologists in standardizing, improving and speeding up their evaluation.

Hong et al. developed a neural network capable of predicting, with a high accuracy, molecular subtypes and 18 common gene mutations in EC, based only on digitalized H&E-stained slides, without the use of sequencing analysis [97]. A similar application might be potentially used as regards predicting MSI and the response to immunotherapy from histological images of ECs. Most of the articles currently available in the literature studied different MSI prediction models in colorectal adenocarcinoma, showing interesting results and good rates of accuracy [98]. In EC, the few published studies used the TCGA dataset for the training, testing, and validation of their MSI prediction models [98].

Zhang et al. introduced a novel deep-learning framework, AMIBA, to predict MSI from digitalized H&E-stained histopathology slides. Their results demonstrated that AI can reliably predict MSI status, relying on the identification of pathologic features, such as immune infiltrate, which are usually recognized by pathologists at microscopical examination [99].

Kather et al. developed a deep residual learning model and tested it on different types of gastrointestinal and endometrial carcinomas. Even though their method showed a robust performance across a range of tumors in predicting molecular features from histology, some limitations were present. One of the limitations was represented by the tissue size. However, the authors stated that biopsies should be sufficient for MSI prediction, even though the evaluation on surgical specimens might show a more robust performance at the moment [100]. Similar results, using different deep-learning models but with the same aim of predicting molecular features from histological images, have been reported by more recent studies from Wang et al. and by Hong et al. [97,101]. All of these data, even though still restricted to few studies and showing some limitations, clearly indicate that AI might potentially represent a helpful and reliable tool in pathologists’ next future for predicting MSI status and other types of molecular alterations in EC.

## 9. Conclusions

MMR/MSI together with *POLE* testing are vital for the accurate histo-molecular classification of EC. Pathologists must become familiar with testing strategies and all their pitfalls. The standardization of technical procedure and interpretation must be the rule in any pathology facility. Some questions still remain to be solved: the intratumoral heterogeneity impact on prognosis and response to ICB; the evaluation of MMR/MSI status in primary vs. metastatic/relapsed endometrial tumors; the prevalence and clinical relevance of secondary MMR loss at metastatic sites; the description of any change in MMR/MSI status after neoadjuvant therapy; and the biological behavior of very rare recently described histological subtypes and their relationship with MMR/MSI and molecular classification.

## Figures and Tables

**Figure 1 ijms-25-01056-f001:**
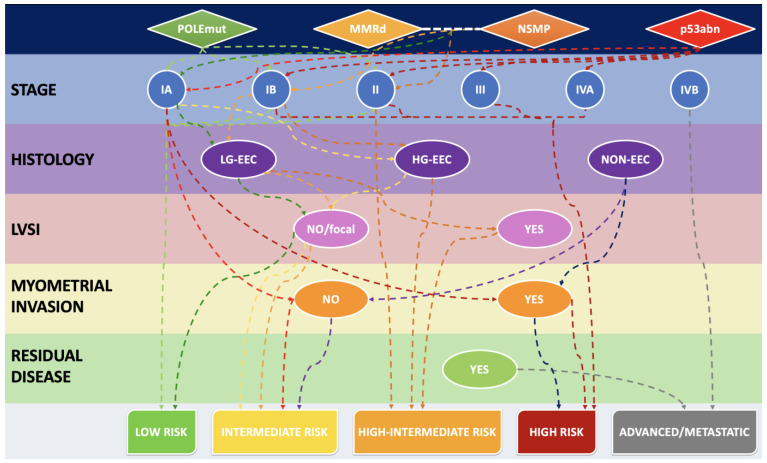
Algorithmic approach to stratify the risk, starting from the molecular group and combining it with staging, histological subtype and other relevant clinicopathological features. LG-EEC: low-grade endometrioid endometrial carcinoma. HG-EEC: high-grade endometrioid endometrial carcinoma. NON-EEC: non-endometrioid endometrial carcinoma.

**Figure 2 ijms-25-01056-f002:**
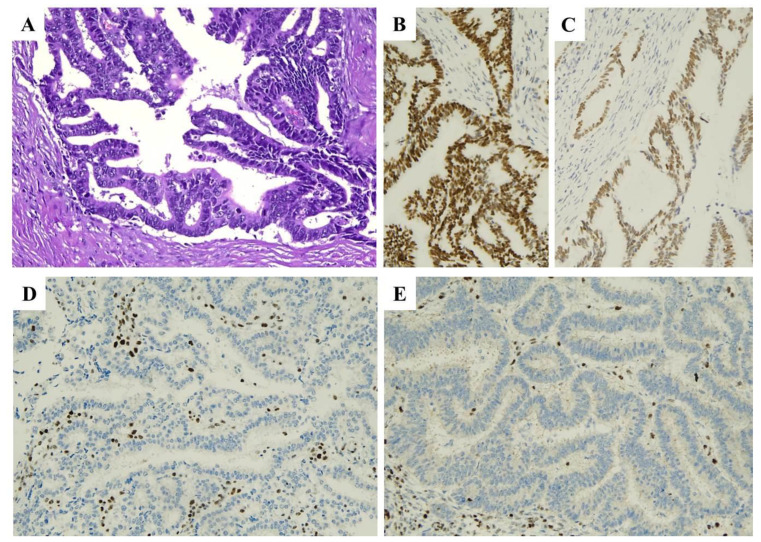
MMR-deficient endometrial carcinoma. (**A**) Haematoxylin and eosin stained section (×20) illustrating an endometrioid carcinoma of moderate grade (G2). (**B**–**E**) By immunohistochemistry, neoplastic cells showed positive staining for MSH6 (**B**) and MSH2 (**C**) and negative staining for MLH1 (**D**) and PMS2 (**E**).

**Figure 3 ijms-25-01056-f003:**
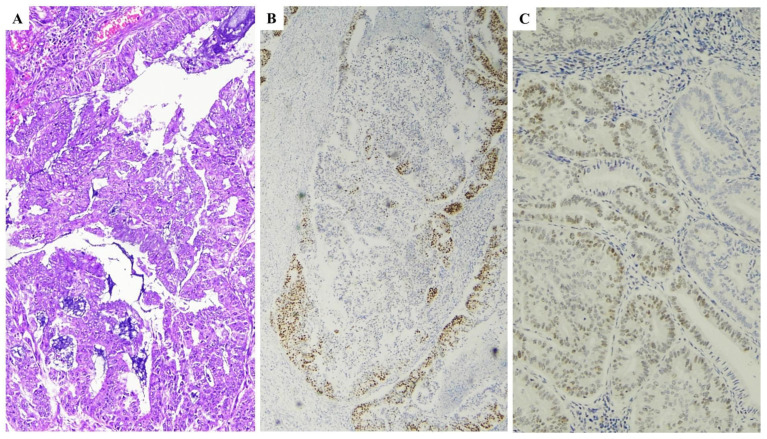
MMR IHC, pitfalls related to poor fixation of the sample. (**A**) Haematoxylin and eosin stained section (×10) illustrating an endometrioid carcinoma of moderate grade with extensive artifacts related to poor fixation. (**B**,**C**) By immunohistochemistry, neoplastic cells showed patchy positivity for MSH2 (**B**) and MLH1 (**C**). Since a valid internal positive control was not found, the case was interpreted as MMR stable.

**Figure 4 ijms-25-01056-f004:**
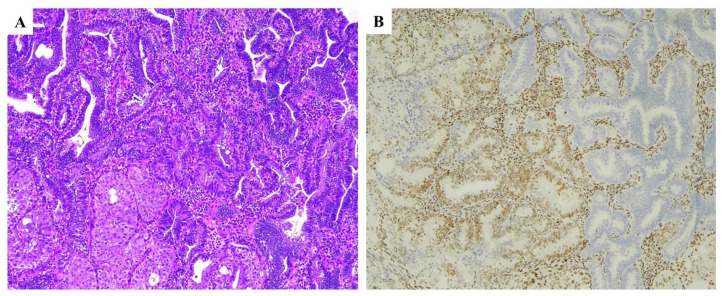
Subclonal MLH1 loss. (**A**,**B**) Haematoxylin and eosin stained section (×20) illustrating an endometrioid carcinoma of moderate grade showing subclonal MLH1 loss, (**B**) involving 10% of the tumor area.

**Figure 5 ijms-25-01056-f005:**
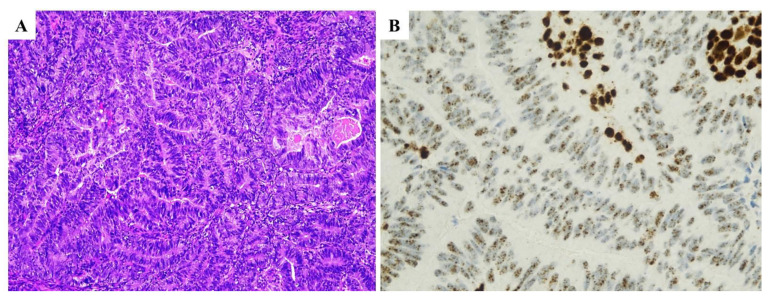
MMR IHC, artefactual, dot-like staining. (**A**,**B**) Haematoxylin and eosin stained section (×20) illustrating an endometrioid carcinoma of moderate grade showing a positive, dot-like staining for MLH1 (**B**) with associated intact nuclear staining in the stromal lymphocytes. Dot-like staining is thought to be a technical artifact which should not be considered as intact expression.

**Table 1 ijms-25-01056-t001:** Histopathological features frequently encountered in MMRd/MSI endometrial carcinoma.

Histopathological Features of MMRd/MSI Endometrial Carcinoma
Lower uterine segment (LUS) origin
Endometrioid differentiation
Severe nuclear atypia with undifferentiated component
High mitotic index
High tumor-infiltrating lymphocytes (TILs) and/or peri-tumoral lymphocytes (≥40 TIL/10HPFs, with more CD8+, CD45RO+ and PD1+ T cells at the invasive tumoral margin)
High morphological heterogeneity
Substantial lympho-vascular space invasion (LVSI)
Deeper myometrial invasion
Synchronous ovarian cancer (clear cell or endometrioid variants)

**Table 2 ijms-25-01056-t002:** Prevalence of different histological types of MMRd/MSI endometrial carcinoma.

Prevalence of Different Histotypes of MMRd/MSI Endometrial Carcinoma
Undifferentiated/dedifferentiated carcinoma (UEC/DEC): **44%**
Neuroendocrine carcinoma: **42.9%**
High-grade endometrioid carcinoma: 39.7%
Mixed: 33.3%
Low-grade endometrioid carcinoma: 24.7%
Clear cell carcinoma: 9.8%
Carcinosarcoma: 7.3%
Serous carcinoma (sporadic)
Mesonephric-like carcinoma (sporadic)

**Table 3 ijms-25-01056-t003:** 2023 FIGO Staging System of endometrial carcinoma, including molecularly defined stages (in blue italic).

*2023 Figo Stage*	*Defining Criteria*
*IA1*	non-aggressive histological type limited to the endometrium or an endometrial polyp
*IA2*	non-aggressive histological type involving <50% myometrium, with no/focal LVSI
*IA3*	low-grade EEC limited to the uterus and ovary
* IAm_POLEmut_ *	* POLEmut EC, confined to the uterine corpus or with cervical extension, regardless of LVSI or histological type *
*IB*	non-aggressive histological type involving ≥50% myometrium, and with no/focal LVSI
*IC*	aggressive histological type limited to the endometrium or an endometrial polyp
*IIA*	non-aggressive histological type with invasion of the cervical stroma
*IIB*	non-aggressive histological type with substantial LVSI
*IIC*	aggressive histological type with any myometrial infiltration
* IICm_p53abn_ *	* p53abn EC, confined to the uterine corpus with any myometrial infiltration, with or without cervical invasion, and regardless of LVSI or histological type *
*IIIA1*	spread to ovary or fallopian tube (except if it meets the Stage IA3 criteria)
*IIIA2*	involvement of uterine subserosa/serosa
*IIIB1*	metastasis or direct spread to the vagina and/or the parametria
*IIIB2*	metastasis to the pelvic peritoneum
*IIIC1*	metastasis to the pelvic lymph nodes (micrometastasis = IIIC1i/macrometastasis = IIIC1ii)
*IIIC2*	metastasis to para-aortic lymph nodes up to the renal vessels, with or without metastasis to the pelvic lymph nodes (micrometastasis = IIIC2i/macrometastasis = IIIC2ii)
*IVA*	invasion of the bladder mucosa and/or the intestinal mucosa
*IVB*	abdominal peritoneal metastasis beyond the pelvis
*IVC*	distant metastasis, including metastasis to any extra- or intra-abdominal lymph nodes above the renal vessels, lungs, liver, brain or bone

**Table 4 ijms-25-01056-t004:** Typical immunohistochemical combinations of MMRd and related MMR molecular defects.

MMR IHC Pattern	MMR Molecular Defect
MLH1+PMS2 loss	*MLH1 promoter hypermethylation*
MLH1+PMS2 loss	*MLH1 gene defect (germline or somatic)*
PMS2 loss	*PMS2 gene defect (germline or somatic)*
MSH2+MSH6 loss	*MSH2 gene defect (germline or somatic)*
MSH6 loss	*MSH6 gene defect (germline or somatic)*

## Data Availability

Not applicable.

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
