# Peer review of "Mismatch Repair Deficiency as a Predictive and Prognostic Biomarker in Endometrial Cancer: A Review on Immunohistochemistry Staining Patterns and Clinical Implications"

_ijms, 2024, doi:10.3390/ijms25021056_

Round 1
Reviewer 1 Report
Comments and Suggestions for Authors
The MS entitled “Mismatch repair deficiency as a predictive and prognostic biomarker in endometrial cancer: a review on immunohistochemistry staining patterns and clinical implications” is a well-written presentation of the current state of the art on the subject.
Major Comments:
1. Please elaborate on the relationship of the Mismatch repair deficiency as a predictive and the relationship of the Mismatch repair deficiency as a prognostic biomarker in endometrial cancers from the viewpoint of different molecular and histological subtypes.
2. Please discuss how the aggressiveness of mismatch repair (MMR) deficient endometrial carcinomas can be revisited in light of different molecular and histological subtypes. A diagrammatic representation will be appreciated by a serious reader.
3. It is known that certain histological subtypes of EC harbor certain types of genomic alterations. A subcategorization of these subtypes in the light of MMR type will make the review novel.
4. The section entitled “Problems of IHC interpretation” should be supported by representative IHC photomicrographs from real-life patients. Authors should present at least one representative IHC photomicrograph for each category as an illustration to make their case. For example, Weak and or focal expression, Poor fixation, cauterization artifacts, neoadjuvant chemotherapy and freezing of tissues, Numerous lymphocytes and stromal cells, Heterogeneous loss (subclonal pattern), Unusual stainings, and Primary vs. Metastasis.
5. A paragraph with a futuristic viewpoint would be appreciated.
Minor Comments:
1. Please provide the name of the author in the place of “some Authors” in line (72)….” According to some Authors [6], an adequate diagnostic algorithm should include immunohistochemical evaluation of MMR”. Also, why is “Authors” capitalized?
2. In the section of …………“Author Contributions: Conceptualization, AS and GFZ; methodology, GA and FI;”…….it is not clear what the methodology implies as there are no figures/data presented in the review.
Comments on the Quality of English Language
Please correct a few typos.
Author Response
The MS entitled “Mismatch repair deficiency as a predictive and prognostic biomarker in endometrial cancer: a review on immunohistochemistry staining patterns and clinical implications” is a well-written presentation of the current state of the art on the subject.
Major Comments:
- Please elaborate on the relationship of the Mismatch repair deficiency as a predictive and the relationship of the Mismatch repair deficiency as a prognostic biomarker in endometrial cancers from the viewpoint of different molecular and histological subtypes.
- Please discuss how the aggressiveness of mismatch repair (MMR) deficient endometrial carcinomas can be revisited in light of different molecular and histological subtypes. A diagrammatic representation will be appreciated by a serious reader.
- It is known that certain histological subtypes of EC harbor certain types of genomic alterations. A subcategorization of these subtypes in the light of MMR type will make the review novel.
We thank the Reviewer 1 for the precious advices that allowed us to significantly improve our manuscript. We believe that these suggestions will make the manuscript of great interest for the readers, thanks to the novel type of information now included in the revised version. As suggested by Reviewer 1, we considerably implemented the text, elaborating in a more detailed manner the relationship of MMR deficiency as predictive and prognostic biomarker, discussing its influence on aggressiveness and the complex correlation between histological subtypes and molecular groups in endometrial cancer.
The chapter 3 of the manuscript has been completely reorganized and subdivided into three paragraphs.
The first one discusses the prognostic/predictive role of MMRd, exploring the relationship between molecular and histological subtypes (as requested in major comment n. 1).
The third paragraph regards the relationship between MMRd and aggressiveness, in the light of the ESGO 2021 and FIGO 2023 updates, regarding a combined histological and molecular approach for risk stratification; moreover, we designed and included an original figure showing a diagrammatic representation of the ESGO 2021 guidelines (as suggested in major comment n. 2) and also inserted a table summarizing the updated FIGO 2023 staging system.
The second paragraph deeply discusses the most common genetic alterations that have been described in each specific histological subtype of EC, also comparing them with some specific alterations more frequently observed in some TCGA molecular subgroups (as suggested in major comment n.3).
For each paragraph, recent relevant articles from the literature have been found, discussed, cited and properly included in the updated list of references.
- The section entitled “Problems of IHC interpretation” should be supported by representative IHC photomicrographs from real-life patients. Authors should present at least one representative IHC photomicrograph for each category as an illustration to make their case. For example, Weak and or focal expression, Poor fixation, cauterization artifacts, neoadjuvant chemotherapy and freezing of tissues, Numerous lymphocytes and stromal cells, Heterogeneous loss (subclonal pattern), Unusual stainings, and Primary vs. Metastasis.
We thank the Reviewer for this suggestion, that we particularly appreciate. We totally agree with her/him, in fact, in the first version we sent to the journal, the figures were included, but the handling editor asked to remove them and make reviewers decide if histological images had to been included in a review or not. We believe, as written by the Reviewer, that representative HE and IHC photographs greatly support the section “Problems of IHC interpretation” and will be particularly appreciated by the readers of the journal. For this reason, in accordance with the Reviewer’s suggestion, we included back the figures with representative HE and IHC images. These figures do not represent novel data coming from scientific experiments. They are only representative and illustrative (unpublished) photomicrographs of real-life problems during histological examination.
- A paragraph with a futuristic viewpoint would be appreciated.
We thank again the Reviewer for the suggestion, that allowed us to include in the last chapter, before conclusions, a paragraph regarding the application of machine learning models and AI to MSI status prediction. We discussed this topic on the basis of the most updated literature, briefly explaining the current results and limitations, the potential help that this tool may give to pathologists and the future perspectives.
Minor Comments:
- Please provide the name of the author in the place of “some Authors” in line (72)….” According to some Authors [6], an adequate diagnostic algorithm should include immunohistochemical evaluation of MMR”. Also, why is “Authors” capitalized?
Amended, thank you.
- In the section of …………“Author Contributions: Conceptualization, AS and GFZ; methodology, GA and FI;”…….it is not clear what the methodology implies as there are no figures/data presented in the review.
GA and FI carefully searched in the literature and selected the relevant articles. Moreover, they helped in designing the figures that have now been included in the revised version of the paper.
Reviewer 2 Report
Comments and Suggestions for Authors
This review focused on MMR deficient ECs. It is a well written manuscript and all relevant literature is covered. In the methodology section the authors should describe which databases have been searched and which search strategies have been used. Study inclusion should be presented in a flowchart. Apart from that I have no specific comments.
Author Response
We greatly thank the Reviewer 2 for the appreciation. As suggested, we included at the end of the Introduction section the methodology that we used and the databases and queries that have been searched. Even though we agree with her/him and really appreciate the suggestion, we did not include a flowchart, because our aim was to provide a narrative review and a practical guide for pathologists in the assessment of MMRd by immunohistochemistry. We did not conduct a systematic review, hence the reason why a methodology section and a study inclusion flowchart are missing in the manuscript.